# A cross-sectional study on COVID-19-related changes in self-medication with antibiotics

**Mohammad Reza Khami[1,2], Armin Gholamhossein Zadeh[3], Dorsa Rahi** [3]*

1 Research Center for Caries Prevention, Dentistry Research Institute, Tehran University of Medical Sciences, Tehran, Iran, 2 Department of Community Oral Health, School of Dentistry, Tehran University of Medical Sciences, Tehran, Iran, 3 Dental Research Center, School of Dentistry, Guilan University of Medical Sciences, Rasht, Iran

* rahi.dorsa@gmail.com

**Data Availability Statement:** All relevant data are within the paper and its Supporting Information files.

## Abstract

### Background and aim

Dental treatments have been limited to emergency care in many countries worldwide due to the global rapid spread of coronavirus disease-2019 (COVID-19). Fear of contracting the disease in dental clinics has also altered the pattern of dental visits and self-medication. The present study compared self-medication with antibiotics (SMA) and the pattern of dental visits before and after the emergence of COVID-19 pandemic in a referral dental clinic in the north of Iran.

### Materials and methods

The data for the present cross-sectional study was collected from 756 patient records retrieved from the archives of the Faculty Clinic of Rasht School of Dentistry during two separate periods: before the COVID-19 pandemic from mid-November 2019 to mid-February 2020, and after the pandemic emergence from mid-April to mid-July 2020. In addition to demographic variables namely age, gender, and place of residence of patients, their smoking status, chief complaint, and SMA were also extracted from patient records. The Chi-square test and binary logistic regression models with 95% confidence interval served for statistical analysis.

### Results

In total, 756 patient records (412 records from the pre-pandemic period and 344 records from the post-pandemic period) were evaluated. SMA was significantly more prevalent after the pandemic compared to that before pandemic (OR = 3.39, 95% CI = 2.43–4.73, P<0.001). The number of patients who smoke significantly decreased after the pandemic by 6.6% compared to that in pre-pandemic period. Dental pain, pus discharge, and abscess as the chief complaints of patients were significantly more prevalent during the post-pandemic period; while, dental checkups, tooth hypersensitivity, and esthetic dental problems were significantly more frequent as the chief complaints of patients during the pre-pandemic period.

**Funding:** all financial support and funding were provided by Tehran University of Medical Sciences (Grant number: 99-3-234-50077).

**Competing interests:** The authors have no conflict of interests to declare.

## Conclusion

There is indication that during the COVID-19 pandemic, SMA and prevalence of acute dental problems in patients have increased. With regard to the consequences of SMA, there is a need to raise public awareness on this matter.

## Introduction

The coronavirus disease-2019 (COVID-19) is an emerging infectious disease, rapidly evolving worldwide [1]. World Health Organization (WHO) announcement on 11th of March 2020 officially declared the spread of the COVID-19 virus as a global pandemic [2]. The disease emerged unexpectedly, and has now turned into a challenging dilemma not only for the public, but also health professionals including dental practitioners, physicians, medical and dental students worldwide [3]. The American Dental Association released the list of emergency and non-emergency dental procedures for dental practitioners and the lay people, and emphasized on limiting the provision of dental services to emergency care only [4]. Such instructions should be strictly followed for prevention and control of COVID-19 [5].

In Iran, all dental offices and clinics were shut down in the first 2 months following the announcement of COVID-19 pandemic by the order of the Iranian Ministry of Health and Medical Education, and only some certain clinics, designated by the Ministry of Health remained open to provide emergency dental services and safety protocols for dental care provision were compiled [6].

In addition to the limitations set by the Ministry of Health, patients less commonly present to dental clinics and offices due to fear of contracting COVID-19. Resultantly, the patient flow and the chief complains of patients have greatly changed since the emergence of COVID-19. A retrospective study conducted in China on 2,537 patients reported a reduction in patient referrals to dental emergency care centers by 38% following the emergence of COVID-19 [7].

According to a definition by the World Health Organization, self-medication refers to medication intake without a prescription, refilling old prescriptions, sharing medications with the family members or one's social cycle, or using the available leftover medications [8, 9]. The reasons for self-medication may include limited access to healthcare facilities, shortage of healthcare services, illegal distribution of medications, wrong beliefs about physicians, and poor knowledge of individuals [10]. Self-medication with antibiotics (SMA) has significant adverse effects such as drug toxicity, resistance of microorganisms, prolonged hospitalization, unsuccessful treatment, high cost of treatment, and increased rate of incurable diseases [11]. In contrast to other medications and almost all other modalities, the efficacy of antibiotics is decreasing over time [12]. A cross-sectional study of self-medication patterns of adults presenting to the dental clinic of Sharjah Dental School for their oral and dental problems revealed that 70.7% of patients had tried self-medication. The most common reason for self-medication was reported to be time shortage for visiting a dentist (37.6%), and not taking oral and dental problems seriously (36.8%) [13]. According to the available statistics, the rate of medication intake in Iran is three times the mean global rate [14, 15].

Considering the increasing use of antibiotics, emergence of multi-drug resistant microorganisms is among the most important public health dilemmas [16, 17]. To the best of our knowledge, no report existed on the COVID-19-related changes among dental patients in Iran. With regard to the alarming consequences of the possible increase in SMA among dental patients during the COVID-19 pandemic, the present study aimed to compare SMA and the

pattern of dental visits before and after the emergence of COVID-19 pandemic in a referral dental clinic in the north of Iran.

## Materials and methods

### Study design and participants

This cross-sectional study evaluated 756 records of patients older than 18 years old, referring to the Faculty Clinic of Rasht School of Dentistry during the morning shifts. The patients had been visited by a general dentist during a 6-month period, starting 3 months before the emergence of COVID-19 pandemic from mid-November 2019 to mid-February 2020, and after the emergence of COVID-19 pandemic from mid-April to mid-July 2020. The patient records were evaluated anonymously, and the extracted data were recorded in datasheets. There is a routine statement in the hospital records, requiring the patients to show if they give consent to use their data anonymously for research purposes. The patients have the option to deny. In our study, all the patients' had shown agreement to use their data for research purposes. This was considered as written informed consent. The study was approved by the Ethics Committee of Tehran University of Medical Sciences (IR.TUMS.DENTISTRY. REC.1399.136).

The minimum sample size was calculated to be 350 for each time period considering 0.29 ratio (prevalence of SMA) in a previous study [18] before the emergence of COVID-19 and 0.42 ratio according to a pilot study, alpha = 0.05, and d = 0.13. The sample size was calculated according to the main objective of the study. From the 2 years ago, a question about SMA has been added to the records of this clinic, and among the routine dental examination, the history of SMA is also recorded in the hospital records of the patients

### Data collection

A general dentist evaluated patient records and retrieved the required information including the patients' gender, age, place of residence (capital city of the province, other urban areas, rural areas), smoking status, chief complaint, and SMA. All available complete records in the specified periods were evaluated, and the required data were entered to the data collection form.

### Statistical analysis

Normal distribution of data was evaluated using the Shapiro-Wilk test, Kolmogorov-Smirnov test, or the kurtosis and skewness measurement. Chi-square test was used to find significant associations between the categorical independent (time before and after the COVID-19 pandemic) and dependent variables. The logistic regression model was fitted to the data to control for the effect of possible confounders. SPSS version 24 (IBM Corp, Armonk, NY, USA) served for statistical analysis at 0.05 level of significance.

## Results

Of 756 patients included in this study, 412 (54.5%) were recruited from the pre-pandemic and 344 (45.5%) were recruited from the post-pandemic period. Of those recruited before the pandemic emergence, 195 (47.3%) were males. The mean age of patients recruited before the pandemic was 39.02 ± 13.84 years. Of those recruited after the pandemic emergence, 160 (46.5%) were males, and the mean age of these patients was 40.38 ± 15.91 years. The patients were homogeneous in terms of gender (p = 0.822) and age (p = 0.217) before and after the COVID-19 pandemic emergence (Table 1).

**Table 1. Comparisons of the studied variables before and after the COVID-19 pandemic emergence among a group of dental patients (n = 756) in Iran.**

| | Time | | P-value ($\chi^2$) |
|---|---|---|---|
| | Before the pandemic n (%) | After the pandemic n (%) | |
| **Gender** | | | |
| **Male** | 195 (47.3) | 160 (46.5) | 0.822 (0.05) |
| **Female** | 217 (52.7) | 184 (53.5) | |
| **Place of residence** | | | |
| **Capital city** | 229 (55.6) | 214 (62.2) | 0.142 (3.90) |
| **Urban areas** | 113 (27.4) | 75 (21.8) | |
| **Rural areas** | 70 (17) | 55 (16) | |
| **Smoking status** | | | |
| **Non-smoker** | 319 (77.4) | 289 (84) | 0.023 (5.16) |
| **Smoker** | 93 (22.6) | 55 (16) | |
| **Chief complaint** | | | |
| **Checkup** | 70 (17) | 7 (2) | <0.001 (115.43) |
| **Pain** | 123 (29.9) | 166 (48.3) | |
| **Pus and abscess** | 93 (22.6) | 124 (36) | |
| **Fracture** | 36 (8.7) | 30 (8.7) | |
| **Hypersensitivity** | 39 (9.5) | 16 (4.7) | |
| **Esthetic problems** | 51 (12.4) | 1 (0.3) | |
| **Self-medication with antibiotics** | | | |
| **No** | 303 (73.5) | 151 (43.9) | <0.001 (68.69) |
| **Yes** | 109 (26.5) | 193 (56.1) | |
| **Medication** | | | |
| **None** | 303 (73.5) | 151 (43.9) | <0.001 (80.57) |
| **Amoxicillin** | 44 (10.7) | 78 (22.7) | |
| **Co-amoxiclav** | 11 (2.7) | 13 (3.8) | |
| **Metronidazole** | 8 (1.9) | 8 (2.3) | |
| **Azithromycin** | 3 (0.7) | 16 (4.7) | |
| **cefixime** | 3 (0.7) | 3 (0.9) | |
| **Penicillin** | 7 (1.7) | 6 (1.7) | |
| **Doxycycline** | 4 (1) | 3 (0.9) | |
| **Clindamycin** | 1 (0.2) | 2 (0.6) | |
| **Incomplete information** | 19 (4.6) | 51 (14.8) | |
| **Amoxicillin & Metronidazole** | 9 (2.2) | 12 (3.5) | |
| **Azithromycin & Penicillin** | 0 (0) | 1 (0.3) | |

Table 1 shows that smoking, chief complaint, SMA, and the type of antibiotics taken varied significantly before and after the pandemic emergence. The prevalence of smoking and the frequency of such chief complaints as dental checkups, tooth hypersensitivity, and esthetic dental problems decreased after the pandemic emergence compared to before that. However, the frequency of dental pain, pus discharge, and abscess as the chief complaints of patients increased after the pandemic emergence (Fig 1). SMA also significantly increased during the pandemic, with Amoxicillin and Azithromycin being the most commonly used antibiotics.

To evaluate the strength of the associations, simple and multiple logistic regression models were fitted to the data (Table 2). According to the adjusted model, patients presented to the clinic after the pandemic were more likely to report SMA compared to those before the pandemic (OR = 3.38, 95% CI = 2.42–2.71, P<0.001). Moreover, having pain (OR = 1.96, 95%

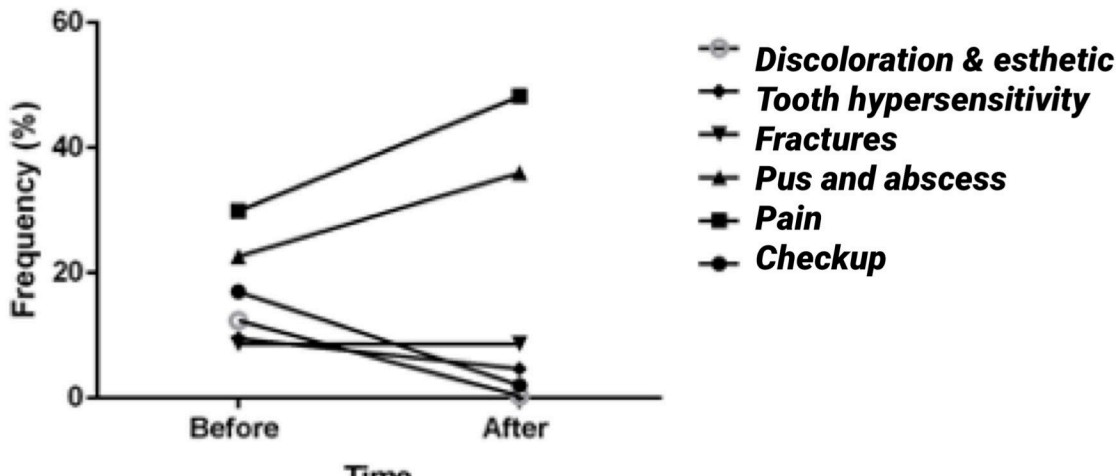

**Fig 1. Changes in the chief complaints of patients after the COVID-19 pandemic emergence compared with before among a group of dental patients (n = 756) in Iran.**

CI = 1.02–3.78, P = 0.04), and pus an abscess (OR = 2.14, 95% CI = 1.09–4.18, P = 0.03) as the chief complaint of the patients was significantly associated with higher SMA report.

## Discussion

The present study investigated SMA and its associated factors among the dental patients before and after the COVID-19 pandemic. According to the results, the rate of SMA increased by 30% after the pandemic emergence, compared to that before. SMA was also associated with the chief complaints of the patients.

Considering the complications and adverse consequences of SMA, and the fact that SMA is a major public health concern, the increase in SMA after pandemic is an alarming finding that calls for attention of dental clinicians and health care professionals, and highlights the need for public awareness on this topic. The increased rate of SMA appears to be due to the postponement of dental visits by patients as the result of fear of contracting COVID-19, leading to self-

**Table 2. Association of smoking, COVID-19 pandemic, and chief complaint with self-medication with antibiotics among a group of dental patients (n = 756) in Iran using simple (unadjusted) and multiple (adjusted) logistic regression models.**

|  | P value | Unadjusted OR | 95% C.I. for OR | | P value | Adjusted OR | 95% C.I. for OR | |
|---|---|---|---|---|---|---|---|---|
|  |  |  | Lower | Upper |  |  | Lower | Upper |
| **COVID-19 pandemic** | <0.001 | 3.55 | 2.62 | 4.82 | <0.001 | 3.38 | 2.42 | 4.71 |
| **Smoking** | 0.089 | 1.39 | 0.95 | 2.02 | 0.29 | 1.24 | 0.83 | 1.84 |
| **Chief complaint*** |  |  |  |  |  |  |  |  |
| Pain | <0.001 | 3.53 | 1.89 | 6.58 | 0.04 | 1.96 | 1.02 | 3.78 |
| Pus and abscess | <0.001 | 3.78 | 1.10 | 7.14 | 0.03 | 2.14 | 1.09 | 4.18 |
| Fractures | 0.04 | 2.25 | 1.04 | 4.87 | 0.40 | 1.41 | 0.63 | 3.17 |
| Tooth hypersensitivity | 0.007 | 3.00 | 1.36 | 6.62 | 0.04 | 2.40 | 1.06 | 5.43 |
| Tooth coloration and esthetic | 0.036 | 2.38 | 1.06 | 5.37 | 0.02 | 2.68 | 1.18 | 6.12 |
|  |  |  |  | **Constant** | <0.001 | 0.16 |  |  |

* Reference: Checkup as the chief complaint

medication. On the other hand, late seeking of dental care would further complicate the problem and increase the rate of emergency cases [4, 19], which was also highlighted in the present study since the results showed a shift in the chief complaints of patients from non-emergency dental problems before the pandemic to emergency problems after the pandemic emergence.

Of self-medicated antibiotics (including Amoxicillin, Co-amoxiclav, Metronidazole, Azithromycin, Cefixime, Penicillin, Doxycycline and Clindamycin), Azithromycin and Amoxicillin had been more commonly used compared to others. Amoxicillin is commonly prescribed by dental practitioners [20]. Self-medication with Azithromycin is probably related to the primary assumptions of patients regarding the effectiveness of this antibiotic against the coronavirus [21].

No significant correlation was noted between SMA and place of residence (urban versus rural areas) or age of patients in the present study. Gender had no significant correlation with SMA before or after the pandemic emergence. Also, the referral rate of male and female patients was almost the same in the present study while Radeva et al. reported higher referral rate of females both before and after the pandemic emergence [22]. Another study on utilization of dental services also reported higher utilization of dental services by females [23].

The chief complaints of patients in our study mainly included dental checkup, esthetic dental problems, tooth discoloration, and tooth hypersensitivity before the pandemic emergence; while, after the emergence of the pandemic, the chief complaints included dental pain, abscess, and tooth fracture. The increase in SMA after the pandemic can be partly explained by the association of SMA with chief complaint of the patients. That the patients with acute problems such as pain, and pus and abscess were more likely to report SMA is not surprising, and similar findings have been reported in Malik et al. [24] and Zhang et al. [25] studies. Moreover, our findings indicate patients sought dental care for more serious and more acute conditions after the pandemic emergence due to the fear of contracting COVID-19 and the set restrictions, which is in line with previous studies [26]. This, in turn, seems to leading more SMA among patients. Anyway, the public should be aware that even for such acute and discomforting problems, and even in pandemic era antibiotics should be prescribed by an authorized health professional [27]. On the other hand, emergency dental care should be accessible for the public during disasters and pandemics [28]. Another reason for increased SMA after pandemic might be the lack of patient information regarding provision of dental care services during the pandemic period, resulting in higher percentage of emergency cases. This calls for sufficient and timely provision of health service information for public during the pandemics and disasters.

In addition to home quarantine, depression, fear, and anxiety related to COVID-19 pandemic have resulted in an increase in the prevalence of hysterical dental pain. Also, home quarantine and greater consumption of sugary substances have aggravated the pattern of dental caries [29].

Our findings highlight the significance of patient education, prevention, and oral health promotion during the pandemics, and dental practitioners should more actively participate in preventive programs and oral health instruction during the COVID-19 pandemic and possible future pandemics.

The present study was carried out in a large professional dental clinic, which is a referral center for all types of oral and dental conditions in a large city in Iran. Also, all the information was extracted and recorded by one dentist, which was another strength of this study.

## Limitations of the study

The cross-sectional design was the main limitation of the present study, as this study type is unable to assess causation. Cross-sectional studies are used to look for the presence or absence

of an outcome as well as the presence or absence of an exposure at a certain period. Because both the outcome and the exposure are investigated at the same time, the temporal link between the outcome and the exposure cannot be ascertained in cross-sectional research [30]. Because a chronological sequence cannot be established, the cross-sectional study cannot be utilized to infer causality. This type of study is still used to obtain descriptive data about a population's disease/outcome burden or to assess background exposure rates [31].

Another limitation was the self-report nature of the study, which induces the risk of recall bias. Moreover, it increases the possibility of giving favorable responses by the respondents, which is called "social desirability" [32]. Thus, the results may be an optimistic estimation of the real situation. In questionnaire research, there's a chance of misunderstandings and errors [33]. That our questionnaire was a part of routine medical and dental records of the patients reduces the likelihood of this bias, however. Anyway, the results should be interpreted cautiously

## Conclusion

There is indication that during the COVID-19 pandemic, SMA and prevalence of acute dental problems in patients have increased. With regard to the consequences of SMA, there is a need to raise public awareness on the subject. Moreover, the public should be informed about the significance of early referral to dentists in order to prevent acute dental problems. The role of dental practitioners is pivotal in this regard.

## Supporting information

**S1 File.**
(DOCX)

**S1 Data.**
(XLSX)

## Author Contributions

**Conceptualization:** Mohammad Reza Khami, Dorsa Rahi.

**Data curation:** Dorsa Rahi.

**Formal analysis:** Armin Gholamhossein Zadeh.

**Methodology:** Mohammad Reza Khami.

**Validation:** Mohammad Reza Khami.

**Writing – original draft:** Dorsa Rahi.

**Writing – review & editing:** Mohammad Reza Khami, Dorsa Rahi.

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
