## [Decision Letter · Decision Letter 0]

16 Jul 2021

PONE-D-21-19078

COVID-19-related changes in the pattern of dental visits and self-medication with antibiotics

PLOS ONE

Dear,

Thank you for submitting your manuscript to PLOS ONE. After careful consideration, we feel that it has merit but does not fully meet PLOS ONE’s publication criteria as it currently stands. Therefore, we invite you to submit a revised version of the manuscript that addresses the points raised during the review process. Please submit your revised manuscript by 15th August 2021. If you will need more time than this to complete your revisions, please reply to this message or contact the journal office at plosone@plos.org. Please include the following items when submitting your revised manuscript:

We look forward to receiving your revised manuscript.

Kind regards,

Muhammad Shahzad Aslam, Ph.D.,M.Phil., Pharm-D

Academic Editor

PLOS ONE

Journal Requirements:

"We acknowledge that all financial support and funding were provided by the Deputy of

Research, International Campus, Tehran University of Medical Sciences."

"This research had no funding source."

Additional Editor Comments (if provided):

Dear authors please correct the issues given by reviewer and resubmit for another round of peer review.

The reviewer requested to include more statistical analysis and improve the methodology of manuscript.

Reviewers' comments:

Reviewer's Responses to Questions

**Comments to the Author**

1. Is the manuscript technically sound, and do the data support the conclusions?

Reviewer #1: Partly

Reviewer #2: No

2. Has the statistical analysis been performed appropriately and rigorously? 

Reviewer #1: No

Reviewer #2: No

3. Have the authors made all data underlying the findings in their manuscript fully available?

Reviewer #1: Yes

Reviewer #2: No

4. Is the manuscript presented in an intelligible fashion and written in standard English?

Reviewer #1: Yes

Reviewer #2: No

5. Review Comments to the Author

Reviewer #1: The current study assessed the impact of the COVID-19 pandemic on the pattern of dental visits and self-medication with antibiotics.

I have some comments which are given below:

Title

The title covers two domains i.e., changes in dental visits, and practice of self-medication with antibiotics. But the results, discussion, and conclusion only cover and discuss the SMA. I will suggest rewriting the title according to the results presented and discussed.

Abstract

The abstract is well written and well presented. Regarding keywords, kindly refer to Mesh terms.

Introduction

The introduction is written well. Kindly limit the introduction to the aims and objectives of the study. Moreover, kindly provide a detail of the rationale of the study, why this study was conducted?.

In the second paragraph, lines between references 5 and 6, the authors were not citing any reference. Kindly cite here a relevant reference.

Method

In the method section, there is great confusion regarding study design. The authors stated that data was retrieved from the hospitals' records in the mentioned specified durations. But the authors have presented this study as cross-sectional rather than a retrospective study. Kindly explain.

If the study was retrieved from the hospital records, then how the authors took informed consent?

Why the authors used sample size calculation based on SMA prevalence?.

In the 2nd line of the statistical analysis, the author stated, “Independent t-test, test and confidence interval”, name the “test”, or remove it.

Results

The results are well presented. But in table 1, numbers with p-values are not clearly mentioned. Have the authors tabulated here chi-square values?

Kindly marked the type of applied test on the study variables and mention it in the table’s legend (Table 1)

Kindly clarify, if the data was retrieved from the hospital records, then how the responses of the study variable i.e., self-practice with antibiotics were recorded?

Discussion and conclusion

The discussion is well written and well compared. Again, the discussion and conclusion only discuss SMA, it doesn’t assess how the pattern of the dental visits was changed.

Reviewer #2: Abstract

• In results, line no. 4 there would be 6.6 rather than 6,6.

• Conclusion should be based on the results of the study. Was the association between antibiotic resistance and SMA assessed in your study?

• P=<0.001? this is not the correct way to write significance.

Introduction

• Introduction seems to lengthy. It is better to make it to-the-point and crisp.

• World health organization should be World Health Organization. There are several other typos which should be corrected throughout the manuscript.

• In the first paragraph, “and” is written repeatedly line 5 & 7.

• In paragraph 3, second last line, it should be complicates rather than complications.

Methodology

• Based on the fact that it is a pre and post COVID-19 study, please explain the study design.

• I will suggest that the study setting should be part of methods section rather than the introduction section.

• How the authors approached the participants to take their written consent specifically for post COVID-19 patients?

• Please explain if the sample size calculation formula is correct for cross sectional studies.

• What was the sampling strategy?

• In data collection section, what about the sampling criteria?

• Please follow the standard format of writing SPSS.

Results

• In the Table 1, what P() column refers to?

• Table 2 is not in the standard format to present result of logistic regression analysis. It is equally not clear from the Table that what is the dependent variable. Further, Table 2 does not justify the use of logistic regression analysis to show the variation in SMA before and after COVID-19 era.

• Overall, the results section needs a major re-write.

Discussion and conclusion

• I would suggest healthcare professionals rather than personnel.

• Paragraph 2, second last line “the” instead of “he.”

• Can you please give the list of self-medicated antibiotics other than amoxicillin and azithromycin?

• Overall, the Discussion part should be improved and elaborated.

• Conclusion section needs improvement and should be coherent with the results. The recommendations made in the conclusion section should be aligned with the results and main objectives of the study.

6. PLOS authors have the option to publish the peer review history of their article (what does this mean?). If published, this will include your full peer review and any attached files.

Reviewer #1: No

Reviewer #2: No

---

## [Author Response · Author response to Decision Letter 0]

13 Oct 2021

Reviewer #1: 

Dear reviewer, we appreciate your time and accurate comments. Your comments help us to improve our manuscript.

1- Title

The title covers two domains i.e., changes in dental visits, and practice of self-medication with antibiotics. But the results, discussion, and conclusion only cover and discuss the SMA. I will suggest rewriting the title according to the results presented and discussed.

Response: The title has now been revised. “COVID-19-related Changes in Self-medication with Antibiotics”. The aim of the study in both abstract and introduction has now been revised to put SMA first (Abstract: last sentence of Background, and page 4, first paragraph) 

2- Abstract

The abstract is well written and well presented. Regarding keywords, kindly refer to Mesh terms.

Response: We greatly appreciate your comment, the keywords are written according to MeSH. “Self-Medication, Anti-Bacterial Agents, Dentistry, COVID-19” 

Introduction

3-The introduction is written well. Kindly limit the introduction to the aims and objectives of the study. Moreover, kindly provide a detail of the rationale of the study, why this study was conducted?.

Response: Thank you for your comment. We revised the introduction to address your comment. The detail of the rationale for the study has now been added (Page 4, first paragraph)

4. In the second paragraph, lines between references 5 and 6, the authors were not citing any reference. Kindly cite here a relevant reference.

Response: There is no sentence without reference between references 5 and 6.

Method

5-In the method section, there is great confusion regarding study design. The authors stated that data was retrieved from the hospitals' records in the mentioned specified durations. But the authors have presented this study as cross-sectional rather than a retrospective study. Kindly explain.

Response: Thank you for your constructive comment. The type of study was revised to cross sectional (Page4, line10).

6- If the study was retrieved from the hospital records, then how the authors took informed consent?

There is a routine statement in the hospital records, requiring the patients to show if they give consent to use their data anonymously for research purposes. The patients have the option to deny. In our study, all the patients’ had shown agreement to use their data for research purposes. This was considered as written informed consent. This has now been added to the manuscript (page:4, line:15-19). 

7- Why the authors used sample size calculation based on SMA prevalence?.

Response: The SMA was the point of focus in our study, which is now more highlighted after the revisions. That’s why we used this variable to calculate sample size.

8- In the 2nd line of the statistical analysis, the author stated, “Independent t-test, test and confidence interval”, name the “test”, or remove it. 

Response: The test was removed (Page5, line12). 

Results

9- The results are well presented. But in table 1, numbers with p-values are not clearly mentioned. Have the authors tabulated here chi-square values?

Response: Thank you. Yes, the Chi square values are tabulated. The p value is reported and the Chi square value is reported in the parenthesis. ¬P-value ( )

10- Kindly marked the type of applied test on the study variables. (Table 1)

Response: The applied test for table 1 was Chi-square test. This has now been added to the Table 1 (Page5, line 11).

11- Kindly clarify, if the data was retrieved from the hospital records, then how the responses of the study variable i.e., self-practice with antibiotics were recorded?

Response: From the 2 years ago, a question about SMA has been added to the records of this clinic, and among the routine dental examination, the history of SMA is also recorded in the hospital records of the patients. This explanation has now been added to the manuscript (Page 5, line1,2,3).

Discussion and conclusion

12. The discussion is well written and well compared. Again, the discussion and conclusion only discuss SMA, it doesn't assess how the pattern of the dental visits was changed.

Response: According to your valuable previous comments, we changed the title, and now the focus of the manuscript is on SMA. The change in dental visit, as a possible justification of change in SMA has been discussed in the discussion. (Page 10,11,12)

Reviewer #2: 

Abstract

With great regards, we kindly appreciate your time and accurate comments. Your comments help us to improve our manuscript. We revised the manuscript according to the comments, as explained below. 

1- In results, line no. 4 there would be 6.6 rather than 6,6.

Response: 6,6 was revised to 6.6 (Page 1, line 20) 

2- Conclusion should be based on the results of the study. Was the association between antibiotic resistance and SMA assessed in your study?

Response: The association between antibiotics resistance and SMA was not assessed in this study, and the conclusion was revised (Page 2, line 4,5,6). 

3- P=<0.001? this is not the correct way to write significance.

Response: The P value was corrected (Page 1, line 19). 

Introduction

4- Introduction seems to lengthy. It is better to make it to-the-point and crisp.

Response: We revised and shortened the introduction (Pages 2,3,4). 

5- World health organization should be World Health Organization.

Response: The correction was done (Page 2 line10). 

6- In the first paragraph, “and” is written repeatedly line 5 & 7.

Response: The corresponded sentences were revised (Page 2, lines 5 and 7). 

7- In paragraph 3, second last line, it should be complicates rather than complications.

Response: Complications was revised to complicates (Page 3, line 5). 

Methodology

8- Based on the fact that it is a pre and post COVID-19 study, please explain the study design.

Response: Thank you for your valuable comment. The study design was revised to cross sectional study (Page 4, line 10). 

9- I will suggest that the study setting should be part of methods section rather than the introduction section.

Response: Thank you for your suggestion. We added the study setting to methods, and removed it from introduction 

10- How the authors approached the participants to take their written consent specifically for post COVID-19 patients?

Response: There is a routine statement in the hospital records, requiring the patients to show if they give consent to use their data anonymously for research purposes. The patients have the option to deny. In our study, all the patients’ had shown agreement to use their data for research purposes. This was considered as written informed consent. (Page 4, line 15-19).

11- Please explain if the sample size calculation formula is correct for cross sectional studies.

 Since our main outcome was the proportion of the patients with SMA, we used the following formula, which uses proportions to calculate sample size in cross sectional studies. We used an estimated proportion of SMA from previous studies.

The formula is available in the attached word file (Response-Reviewer 2), as it was not possible to copy it here.

12- What was the sampling strategy?

 Any patient referring to the Faculty Clinic of Rasht School of Dentistry during the morning shifts and was older than 18 years old was included in the study.” (Page 4, line 10,11) 

13- In data collection section, what about the sampling criteria?

Response: Inclusion criteria is added to the method. “Any patient referring to the Faculty Clinic of Rasht School of Dentistry during the morning shifts and was older than 18 years old was included in the study.” (Page 4, line10,11, and page 5, lines 8 and 9) 

14- Please follow the standard format of writing SPSS.

Response: Revised to “SPSS version 24 (IBM Corp, Armonk, NY, USA)”

Results

15- In the Table 1, what P() column refers to?

Response: The P refers to p value and X2 refers to the Chi-square value. 

16- Table 2 is not in the standard format to present result of logistic regression analysis. It is equally not clear from the Table that what is the dependent variable. Further, Table 2 does not justify the use of logistic regression analysis to show the variation in SMA before and after COVID-19 era. 

Response: Thank you for your valuable comment. The table 2 was revised substantially. Now the dependent variable is SMA, and the data presentation format has been revised (Pages 9,10).

17-Overall, the results section needs a major re-write.

The results part has now been revised (Pages 9,10).

Discussion and conclusion

18- I would suggest healthcare professionals rather than personnel.

Response: The healthcare professional was changed to personnel (Page10, line9). 

19- Paragraph 2, second last line “the” instead of “he.”

Response: Revised. We kindly appreciate your accuracy. 

20- Can you please give the list of self-medicated antibiotics other than amoxicillin and azithromycin?

Response: Revised to “Of self-medicated antibiotics (including Amoxicillin, Co-amoxiclav, Metronidazole, Azithromycin, Cefixime, Penicillin, Doxycycline and Clindamycin), Azithromycin and Amoxicillin had been more commonly used compared to others” (Page11, line3,4).

21- Overall, the Discussion part should be improved and elaborated

Response: The Discussion part has now been revised and elaborated substantially, with the main focus on SMA (Pages:10,11,12)

22- Conclusion section needs improvement and should be coherent with the results. The recommendations made in the conclusion section should be aligned with the results and main objectives of the study.

Response: We revised Conclusion part to address this comment (Page13, line1-5).

---

## [Decision Letter · Decision Letter 1]

23 Dec 2021

PONE-D-21-19078R1COVID-19-related Changes in Self-medication with AntibioticsPLOS ONE

Dear,

Thank you for submitting your manuscript to PLOS ONE. After careful consideration, we feel that it has merit but does not fully meet PLOS ONE’s publication criteria as it currently stands. Therefore, we invite you to submit a revised version of the manuscript that addresses the points raised during the review process.

We look forward to receiving your revised manuscript.

Kind regards,

Muhammad Shahzad Aslam, Ph.D.,M.Phil., Pharm-D

Academic Editor

PLOS ONE

Journal Requirements:

Reviewers' comments:

Reviewer's Responses to Questions

**Comments to the Author**

1. If the authors have adequately addressed your comments raised in a previous round of review and you feel that this manuscript is now acceptable for publication, you may indicate that here to bypass the “Comments to the Author” section, enter your conflict of interest statement in the “Confidential to Editor” section, and submit your "Accept" recommendation.

Reviewer #1: All comments have been addressed

Reviewer #3: (No Response)

2. Is the manuscript technically sound, and do the data support the conclusions?

Reviewer #1: Yes

Reviewer #3: Yes

3. Has the statistical analysis been performed appropriately and rigorously? 

Reviewer #1: Yes

Reviewer #3: Yes

4. Have the authors made all data underlying the findings in their manuscript fully available?

Reviewer #1: Yes

Reviewer #3: Yes

5. Is the manuscript presented in an intelligible fashion and written in standard English?

Reviewer #1: Yes

Reviewer #3: Yes

6. Review Comments to the Author

Reviewer #1: The authors have addressed all of my comments/suggestions in the revised manuscript.

Reviewer #3: Suggest

1. the add "A cross-sectional study on............." to the title to give a reader the full picture.

2. rewording of some parts of the write-up to provide clarity (pl refer to the reviewed article)

3. Details of Data Collection Form to be included in the write-up.

4. There is a mis-match of the number of respondents - 350 required and only 344 studied?

5. Table 1 and Figure 1 descriptions to be separated.

7. PLOS authors have the option to publish the peer review history of their article (what does this mean?). If published, this will include your full peer review and any attached files.

Reviewer #1: No

Reviewer #3: **Yes: **Prof. Datuk Dr. Allan Mathews

---

## [Author Response · Author response to Decision Letter 1]

5 Mar 2022

We would like to cordially thank the reviewer for the constructive comments. Here is our point by point responses to the comments 

1. the add "A cross-sectional study on............." to the title to give a reader the full picture.

Response: The title was revised according to the comment.

2. rewording of some parts of the write-up to provide clarity (pl refer to the reviewed article)

Response: All requested rewordings were done.

3. Details of Data Collection Form to be included in the write-up.

Response: A sentence on data collection form was added to the paragraph on data collection (page 5). The data collection form included the variables explained in the same paragraph.

4. There is a mis-match of the number of respondents - 350 required and only 344 studied?

Response: We collected the data from all available complete records in the specified time periods. The number of these records in the post-pandemic period was 344. The biostatistician author confirmed that this will not affect the analyses and results.

5. Table 1 and Figure 1 descriptions to be separated.

Response: The results part was revised to address this comment (page 8, first paragraph).

Comments inserted in the text:

Suggest reword - There is indication that during the Covid-19 pandemic, SMA and prevalence of acute dental problems. 

Response: The conclusion in the abstract and in the main text was revised according to the comment.

“patients who smoke” in the results of abstract.

Response: The results of abstract was revised.

Maintain consistency and use Antibiotics (in the keywords).

Response: Revised

“complicates” (in the introduction)

Response: Revised.

Need this be in capitals?? (the Ethics Code)

Response: Yes. The Ethics Approval Codes issued by Tehran University of Medical Sciences are all in this format. It might be better to keep it in capital.

Restate to reflect true meaning (the statistical analysis section)

Response: We revised the paragraph on statistical analysis to reflect true meaning (page 5, last paragraph). We hope it is quite clear now.

antibiotics taken

Response: Revised (page 8, first paragraph).

after the emergence of the pandemic (in discussion).

Response: Revised (page 12, first paragraph).

Check spacing of paragraphs (in discussion part).

Response: Checked and revised.

---

## [Decision Letter · Decision Letter 2]

21 Mar 2022

PONE-D-21-19078R2A Cross-sectional Study on COVID-19-related Changes in Self-medication with AntibioticsPLOS ONE

Dear Dr. Rahi,

Thank you for submitting your manuscript to PLOS ONE. After careful consideration, we feel that it has merit but does not fully meet PLOS ONE’s publication criteria as it currently stands. Therefore, we invite you to submit a revised version of the manuscript that addresses the points raised during the review process. Please include limitation of study and future recomendation as separate heading and explain in details.

We look forward to receiving your revised manuscript.

Kind regards,

Muhammad Shahzad Aslam, Ph.D.,M.Phil., Pharm-D

Academic Editor

PLOS ONE

Journal Requirements:

Additional Editor Comments:

According to author "The present study was carried out in a large professional dental clinic, which is a referral center

for all types of oral and dental conditions in a large city in Iran. Also, all the information was

extracted and recorded by one dentist, which was another strength of this study. However, the

cross-sectional design was the main limitation of the present study"

I request to provide a separate heading under the limitation of study and write my in detail about the limitation of the study

---

## [Author Response · Author response to Decision Letter 2]

19 Apr 2022

Additional Editor Comments:

Comment: According to author "The present study was carried out in a large professional dental clinic, which is a referral center

for all types of oral and dental conditions in a large city in Iran. Also, all the information was

extracted and recorded by one dentist, which was another strength of this study. However, the

cross-sectional design was the main limitation of the present study"

I request to provide a separate heading under the limitation of study and write my in detail about the limitation of the study

Response: We thank the editor for this comment. A separate heading and paragraph about the limitations of the study was added according to the comment (page 13, paragraph 2).

---

## [Editor Report · Decision Letter 3]

28 Apr 2022

PONE-D-21-19078R3A Cross-sectional Study on COVID-19-related Changes in Self-medication with AntibioticsPLOS ONE

Dear Dr. Rahi,

Thank you for submitting your manuscript to PLOS ONE. After careful consideration, we feel that it has merit but does not fully meet PLOS ONE’s publication criteria as it currently stands. Therefore, we invite you to submit a revised version of the manuscript that addresses the points raised during the review process. Please submit your revised manuscript by 28th May 2022. If you will need more time than this to complete your revisions, please reply to this message or contact the journal office at plosone@plos.org. Please include the following items when submitting your revised manuscript:A rebuttal letter that responds to each point raised by the academic editor and reviewer(s). You should upload this letter as a separate file labeled 'Response to Reviewers'.A marked-up copy of your manuscript that highlights changes made to the original version. You should upload this as a separate file labeled 'Revised Manuscript with Track Changes'.An unmarked version of your revised paper without tracked changes. You should upload this as a separate file labeled 'Manuscript'.If applicable, we recommend that you deposit your laboratory protocols in protocols.io to enhance the reproducibility of your results. Protocols.io assigns your protocol its own identifier (DOI) so that it can be cited independently in the future. For instructions see: https://journals.plos.org/plosone/s/submission-guidelines#loc-laboratory-protocols. Additionally, PLOS ONE offers an option for publishing peer-reviewed Lab Protocol articles, which describe protocols hosted on protocols.io. Read more information on sharing protocols at https://plos.org/protocols?utm_medium=editorial-email&utm_source=authorletters&utm_campaign=protocols.

We look forward to receiving your revised manuscript.

Kind regards,

Muhammad Shahzad Aslam, Ph.D.,M.Phil., Pharm-D

Academic Editor

PLOS ONE

Journal Requirements:

Additional Editor Comments:

Please write in detail the limitation of the study design, methodological limitation

Kindly explain why we cannot able to assess causation and state recommendation.

Kindly explain self-report nature of the study and state recommendation.

Do this study have Prevalence-incidence bias? If yes then explain.

Please indicate potential bias in the study and explain in details
---

## [Author Response · Author response to Decision Letter 3]

27 May 2022

Response to Reviewers

Additional Editor Comments:

Comment: Additional Editor Comments:

1. Please write in detail the limitation of the study design, methodological limitation

Kindly explain why we cannot able to assess causation and state recommendation.

Kindly explain self-report nature of the study and state recommendation.

Response: We thank the editor for this comment. The paragraph about the limitations of the study was revised according to the comment, and the requested aspects were covered (page 13, paragraph 2).

“The cross-sectional design was the main limitation of the present study, as this study type is unable to assess causation. Cross-sectional studies are used to look for the presence or absence of an outcome as well as the presence or absence of an exposure at a certain period. Cross-sectional studies can be conducted without the requirement for follow-up, making them more cost-effective. Because both the outcome and the exposure are investigated at the same time, the temporal link between the outcome and the exposure cannot be ascertained in cross-sectional research (30). Because a chronological sequence cannot be established, the cross-sectional study cannot be utilized to infer causality. This type of study is still used to obtain descriptive data about a population's disease/outcome burden or to assess background exposure rates (31).

Another limitation was the self-report nature of the study, which induces the risk of recall bias. Moreover, it increases the possibility of giving favorable responses by the respondents, which is called “social desirability” (32). Thus, the results may be an optimistic estimation of the real situation. In questionnaire research, there's a chance of misunderstandings and errors (33). That our questionnaire was a part of routine medical and dental records of the patients reduces the likelihood of this bias, however. Anyway, the results should be interpreted cautiously.”

2. Do this study have Prevalence-incidence bias? If yes then explain.

Please indicate potential bias in the study and explain in details

Response: We thank the editor for raising this point. Prevalence-incidence bias occurs when individuals with severe or mild disease are excluded, leading to an error in the estimated association between an exposure and an outcome. It seems that we do not have prevalence-incidence bias in our study because our clinic was a referral center and patients came in with any degree of conditions.

---

## [Editor Report · Decision Letter 4]

31 May 2022

A Cross-sectional Study on COVID-19-related Changes in Self-medication with Antibiotics

PONE-D-21-19078R4

Dear,

We’re pleased to inform you that your manuscript has been judged scientifically suitable for publication and will be formally accepted for publication once it meets all outstanding technical requirements.

Kind regards,

Muhammad Shahzad Aslam, Ph.D.,M.Phil., Pharm-D

Academic Editor

PLOS ONE
---

## [Editor Report · Acceptance letter]

6 Jun 2022

PONE-D-21-19078R4 

A Cross-sectional Study on COVID-19-related Changes in Self-medication with Antibiotics 

Dear Dr. Rahi:

I'm pleased to inform you that your manuscript has been deemed suitable for publication in PLOS ONE. Congratulations! Your manuscript is now with our production department. 

Kind regards, 

on behalf of

Dr. Muhammad Shahzad Aslam 

Academic Editor

PLOS ONE